# A Theranostic Nanocomplex Combining with Magnetic Hyperthermia for Enhanced Accumulation and Efficacy of pH-Triggering Polymeric Cisplatin(IV) Prodrugs

**DOI:** 10.3390/ph15040480

**Published:** 2022-04-14

**Authors:** Yang Qu, Zhiqi Wang, Miao Sun, Tian Zhao, Xuanlei Zhu, Xiaoli Deng, Man Zhang, Ying Xu, Hongfei Liu

**Affiliations:** 1College of Pharmacy, Jiangsu University, Zhenjiang 212013, China; quyang@ujs.edu.cn (Y.Q.); 2211915014@stmail.ujs.edu.cn (Z.W.); 2222015024@stmail.ujs.edu.cn (M.S.); 3191603010@stmail.ujs.edu.cn (X.Z.); 3181603070@stmail.ujs.edu.cn (X.D.); 3181603068@stmail.ujs.edu.cn (M.Z.); 2Chia Tai Tianqing Pharmaceutical Group Co., Ltd., Nanjing 210046, China; 3Department of Medical Imaging, Affiliated Hospital of Jiangsu University, Jiangsu University, Zhenjiang 212001, China; locklove@ujs.edu.cn; 4Jiangsu Sunan Pharmaceutical Group Co., Ltd., Zhenjiang 212400, China

**Keywords:** polymeric prodrug, theranostic nanocomplex, pH-triggering releasing, magnetic targeting, magnetic hyperthermia, magnetic resonance imaging

## Abstract

Although polymeric platinum(IV) (Pt(IV)) prodrugs can reduce the side effects of cisplatin, the efficacy of the prodrug is still limited by its non-targeted distribution, poor penetration in deep tumor tissue, and low cytotoxicity in tumor cells. To improve the clinical potential of polymeric prodrug micelle, we synthesized amphiphilic polymeric Pt(IV) with high Pt content (22.5%), then developed a theranostic nanocomplex by integrating polymeric Pt(IV) with superparamagnetic Mn_0.6_Zn_0.4_Fe_2_O_4_ via simple self-assembly. Due to the high content of Mn_0.6_Zn_0.4_Fe_2_O_4_ (41.7% *w*/*w*), the theranostic nanocomplex showed high saturation magnetization (103.1 emu g^−1^) and excellent magnetocaloric effect (404 W g^−1^), both of them indicating its advantages in efficient magnetic targeting (MT), magnetic hyperthermia (MH), and magnetic resonance imaging (MRI). In vitro, in combination with MH, the theranostic nanocomplex showed as high cytotoxicity as cisplatin because of a significant increase in platinum of cellular uptake. In vivo, the accumulation of theranostic nanocomplex in tumors was increased by MT and confirmed by MRI. Furthermore, MH improved penetration of theranostic nanocomplex in tumors as expanding blackened area in tumors was observed by MRI. Based on these properties, the theranostic nanocomplex, under the assistance of MT and MH, showed the highest tumor growth inhibition rate (88.38%) after different treatments, while the body weight of mice increased slightly, indicating low side effects compared to those of cisplatin. The study provided an advanced theranostic nanocomplex with low toxicity and high efficacy, indicating a great clinical potential of polymeric Pt(IV).

## 1. Introduction

Cisplatin, a classical platinum (Pt) drug with +2 valence (Pt(II)), is one of the most successful antitumor drugs against tumor cells by crosslinking DNA via coordination bond; however, its severe side effects are an inevitable problem in clinics [1]. To overcome the shortcoming of cisplatin, several kinds of polymer-based drug delivery systems have been developed to improve cisplatin pharmacokinetics and enhance its accumulation in tumor sites by enhanced permeability and retention (EPR) effect [2]. As cisplatin is a micro-molecule lacking sufficient hydrophobicity and hydrophilicity, physical encapsulation is not suitable for preparing cisplatin-loaded nanocarrier direct. Consequently, a kind of Pt drug with +4 valence (Pt(IV)) has been developed by modifying cisplatin on its axial directions to improve its application potential [3]. The current prevailing view considers the Pt(IV) as a prodrug of cisplatin because its cytotoxicity should be suppressed in blood circulation, then activated by reducing to cisplatin under the effect of intracellular reductants, such as ascorbic acid and glutathione (GSH) [3]. Although Pt(IV) with high hydrophobicity could be encapsulated into micelles by hydrophobic effect easily [4], the physical trapping was unstable, as many studies displayed obvious drug releasing from the nanocarriers under physiological conditions in 24 h, implying the risk of drug leak from corresponding delivery in blood circulation [5,6].

To raise the stability of the Pt(IV)-loaded nanocarrier, an attractive approach is to prepare polymeric prodrug micelle, which immobilizes Pt(IV) on polymers by stimuli-responsive covalent bond and forms prodrug micelle [2]. Because covalent bonding possesses high stability in physical conditions and rapid cleavage in a tumor’s extracellular or intracellular environment, polymeric Pt(IV) micelle shows double advantages on long-circulation in blood and targeted accumulation in the tumor. Moreover, high drug content (≥10 wt.%) in prodrug micelle has been achieved by the optimization of the composition and architecture of polymeric prodrug [7,8]. Therefore, the polymeric prodrug micelle should be a competitive option in the clinical application of cisplatin.

However, the high stability of polymeric prodrug micelle is a double-edged sword. In blood circulation, the high binding energy between polymer and Pt(IV) reduced drug leaking, resulting in low side effects. Although the EPR effect has been involved in many studies to improve the accumulation of drug-loaded nanocarriers in tumors, its effect has been limited in small animal models [9]. So far, drug-loaded nanocarriers have almost uniformly failed to improve efficacy in clinics [10]. Much basic research on tumors has found that diffusion of nanoparticles in tumors has been hindered by some barriers, including tortuous tumor blood vessels, high-density extracellular matrix, and consequent elevated interstitial pressure [11]. Meanwhile, differences between tumor and normal tissue on pH [12] and GSH concentration [13] were insufficient for the breakage of covalent bonding efficiently. Therefore, after polymeric Pt(IV) prodrug accumulated in the tumor site by EPR effect, its penetration into deep tumor was more difficult than that of micro-molecular drugs, such as Pt(IV) and cisplatin, which reduced its efficacy in clinics.

In order to overcome the obstacle, hyperthermia should be a promising strategy to promote the permeation of polymeric prodrug micelle. Previous studies have confirmed that regional mild hyperthermia (40–43 °C) can increase tumor blood supply and accelerate the diffusion of macromolecules in solid tumors [14,15]. In addition, hyperthermia can enhance cellular uptake and cytotoxicity of chemotherapeutic drugs significantly, especially for cisplatin and its derivatives. As hyperthermia can interfere reparation of DNA damages caused by platinum-based antitumor drugs [16], the integration of hyperthermia and polymeric Pt(IV) micelles should be an ideal strategy for improving the efficacy of cisplatin in clinics.

According to progress in noninvasive hyperthermia, photothermal treatment and magnetic hyperthermia (MH) have attracted great attention. However, MH possesses a greater clinic potential because an alternating magnetic field (AMF) for MH can penetrate the human body without any loss, in contrast with depth-limited near-infrared ray (NIR) for photothermal treatment [17]. In our previous study, we developed an MH composed of superparamagnetic Mn_0.6_Zn_0.4_Fe_2_O_4_ nanoparticles with a high specific adsorption rate (SAR) under AMF and confirmed its efficiency on tumor suppression alone [18] and multi-treatment in combination with chemotherapy [19]. Moreover, magnetic fluid consisting of superparamagnetic nanoparticles (SPIO) has also been used widely in fields of magnetic resonance imaging (MRI) [20] and magnetic targeting (MT) [21]. Therefore, SPIO should be an ideal nanomaterial to combine with polymeric Pt(IV) prodrug, obtaining a versatile nanocomplex for diagnosis and treatment of tumors.

In the study, a theranostic nanocomplex was constructed to promote the clinical potential of polymeric Pt(IV) prodrug by self-assembly of monodispersed Mn_0.6_Zn_0.4_Fe_2_O_4_ and amphiphilic polymeric Pt(IV), as shown in Figure 1.

To prepare the theranostic nanocomplex, we synthesized polymeric prodrugs with high platinum content firstly. A hydrophilic diblock copolymer monomethoxyl poly(ethylene glycol)-block-poly(2-hydroxyethyl methacrylate) (mPEG-*b*-pHEMA) (shown in Figure 1) with suitable hydroxide radical content was synthesized by reversible addition–fragmentation chain-transfer polymerization (RAFT), then a considerable amount of modified Pt(IV) with single carboxyl (Pt(IV)-COOH) combined with hydroxyls of PEG-*b*-pHEMA by esterification reaction, resulting in amphiphilic polymeric Pt(IV). By self-assembly, amphiphilic polymeric Pt(IV) could encapsulate hydrophobic monodispersed Mn_0.6_Zn_0.4_Fe_2_O_4_ easily and form the stable micellar theranostic nanocomplex (MTN). MTNs showed advantages in suitable diameter for the EPR effect, high stability under physiological conditions, and obvious acid-triggering drug release. Because of the existence of clustering monodispersed Mn_0.6_Zn_0.4_Fe_2_O_4_, the MTN displayed multipotentials on MRI, MT, and MH, also shown in Figure 1. In accordance with relevant properties, cytotoxicity and cellular uptake of MTN in vitro have been enhanced significantly by efficient MH. Furthermore, in vivo studies showed that accumulation of MTN in tumors was enhanced by MT and observed by MRI. What is more, MH of 20 min promoted permeation of MTN in tumors, which was also confirmed by MRI. As the combination of MT and MH benefited accumulation and penetration of the nanocomplex in tumors, as shown in Figure 1, the nanocomplex exhibited high antitumor efficacy and low side effects because of the slight gaining weight of mice, which increased cisplatin potential in clinics.

## 2. Results and Discussion

### 2.1. Synthesis and Characterization of Polymeric Pt(IV)

The synthetic route of polymeric Pt(IV) is shown in Appendix A, containing two parts, preparation of Pt(IV)-COOH and synthesis of mPEG-*b*-pHEMA. According to previous studies on Pt(IV) [8,22], we prepared Pt(IV) with two hydroxyl groups (Pt(IV)-(OH)*_2_*) at Pt axial positions by oxidization of excessive hydrogen peroxide on cisplatin, then modified the Pt(IV) with a single carboxyl group (Pt(IV)-COOH) by reacting with equimolar succinic anhydride (SA) under mild conditions, also shown in Appendix A. Furthermore, we characterized the structure and molecular weight of Pt(IV)-COOH by Fourier transform infrared (FT-IR) spectrum, H proton nuclear magnetic resonance (^1^H NMR), and mass spectrum, as shown in Appendix A. Compared to relative results in previous studies [22,23,24,25], Pt(IV)-COOH has been prepared successfully.

At the same time, we synthesized diblock hydrophilic polymer, mPEG-*b*-pHEMA, by a combination of esterification and RAFT reactions, which was also displayed in Appendix A. In the reaction, S-1-Dodecyl-S′-(α,α′-dimethyl-α″-acetic acid)trithiocarbonate (DDAT) was selected as chain-transfer agent (CTA) for RAFT polymerization and synthesized according to the literature [26]. Then, we prepared macro-CTA by DCC reaction between monomethoxyl poly(ethylene glycol) (mPEG) and DDAT, resulting in mPEG-DDAT. Its structure was characterized by ^1^H NMR and is shown in Appendix A. After the successful synthesis of macro-CTA, the successive reaction was to synthesize mPEG-*b*-pHEMA with repeating units of 2-hydroxyethyl methacrylate (HEAM) by RAFT reaction because hydroxide radical of HEMA could be used to combine with Pt(IV)-COOH. To figure out whether the RAFT reaction was successful or not, we characterized the structure of mPEG-*b*-pHEMA by ^1^H NMR, as shown in Figure 2 (bottom spectrum).

Compared to a previous relative study [27], all characteristic peaks of pHEMA were observed and marked in Figure 2, which confirmed the successful polymerization of pHEMA as a product of mPEG-DDAT chain propagation. Moreover, the average polymerization degree of HEMA was calculated to be 27, resulting from an integral area ratio of peaks between 3.38 ppm (labeled as 1, terminal C*H*_3_ of mPEG) and 4.7–4.9 ppm (labeled as 8, O*H* of pHEMA).

Based on the structures of Pt(IV)-COOH and mPEG-*b*-pHEMA, we prepared polymeric Pt(IV) by esterification between carboxyl of Pt(IV) and hydroxy of mPEG-*b*-pHEMA. Although most previous studies used N,N′-Dicyclohexylcarbodiimide (DCC) reaction to prepare polymer-Pt(IV) conjugation [8,23,25,26,28,29], we modified the reaction process by a combination of 1-(3-Dimethylaminopropyl)-3-ethylcarbodiimide hydrochloride (EDC) and 1-Hydroxybenzotriazole (HOBt) to simplify post-treatment during product purification. The structure of polymeric Pt(IV) was also characterized by ^1^H NMR, as shown in Figure 2 (top spectrum). Compared to the counterpart of mPEG-*b*-pHEMA, polymeric Pt(IV) not only retained all characteristic peaks of mPEG-*b*-pHEMA but also presented characteristic peaks of Pt(IV) in the ^1^H NMR spectrum of polymeric Pt(IV), which was marked in Figure 2. In contrast with that of the bottom spectrum, the intensity of the peak at 4.7–4.9 ppm (labeled as 8) in the top spectrum reduced obviously, which indicated a decrease in the hydroxy amount of polymeric Pt(IV). Meanwhile, the conversion from hydroxy to ester bond was confirmed by increasing peak intensity at 3.83–3.93 (labeled as 5 + 6′) because C*H*_2_ (labeled as 6′) possessed the same chemical environment as C*H*_2_ (labeled as 5) after forming ester bonding. Furthermore, the Pt content was determined precisely by an inductively coupled plasma mass spectrometer (ICP-MS), which was as high as 22.5 wt%. On the basis of the above results, polymeric Pt(IV) with high Pt content was prepared successfully.

### 2.2. Characterization of Polymeric Pt(IV)-Based Micelles

Using polymeric Pt(IV), we prepared polymeric Pt(IV) micelle (PPM) and MTN; both of them were observed directly by transmission electron microscopy (TEM), as shown in Figure 3A,B, and we further measured their size distributions by dynamic light scattering (DLS), as shown in Figure 3C.

By self-assembly of polymeric Pt(IV), PPM showed a spherical inner core with high contrast without phosphotungstic acid staining, which implied high content Pt of polymeric Pt(IV). Because of the amphipathy of polymeric Pt(IV), MTN could be prepared to encapsulate Mn_0.6_Zn_0.4_Fe_2_O_4_ nanoparticles by the same procedure. Besides PPM and MTN, we also provided a TEM photo of monodispersed Mn_0.6_Zn_0.4_Fe_2_O_4_ nanoparticles, which was shown in the inset of Figure 3B and Appendix A. Apparently, MTN exhibited a larger particle size than that of PPM, as MTN possessed a core of Mn_0.6_Zn_0.4_Fe_2_O_4_ nanocluster. Therefore, we determined the size distributions of PPMs and MTNs in PBS by DLS. As monodispersed Mn_0.6_Zn_0.4_Fe_2_O_4_ nanoparticles displayed high hydrophobicity, their diameter was measured in THF. The size distributions of SPIO, PPM, and MTN increase were 9.8, 86, and 151 nm, respectively (Figure 3C), which were consistent with their TEM results. In order to estimate the stability of MTN, the time-dependent hydrodynamic diameter was studied in PBS further, which was shown in Appendix A, confirming their high stability in physiological buffer. These results not only prove the formation of MTN directly but also suggest the MTN with some potential in clinics, such as suitable size for passive targeting (EPR effect, 10–200 nm) and magnetic properties on magnetic targeting and hyperthermia, because of the existence of highly compacted SPIO cluster in the core of MTN.

After quantifying the content of SPIO in MTN by ICP-MS, the content of SPIO in MTN was calculated as 41.7% *w*/*w*, which corresponded to a feed ratio of SPIO in a gross mass of 50%. Because of the existence of SPIO, as expected, MTN displayed excellent magnetism, as shown in Figure 4A.

According to our previous study [18], monodispersed Mn_0.6_Zn_0.4_Fe_2_O_4_ nanoparticles exhibited high saturation magnetism (*M_s_* = 74.6 emu g^−1^) and superparamagnetism, which reappeared in this research, as shown in Appendix A (*M*_s_ = 102.7 emu g^−1^
_[Fe+Mn+Zn]_, corresponding to 74.6 emu g^−1^ _[mass of SPIO]_). Due to high *M*_s_ and superparamagnetism of Mn_0.6_Zn_0.4_Fe_2_O_4_ nanoparticles, MTN displayed similar hysteresis loops, as shown in Figure 4A, indicating its high *M*_s_ (103.1 emu g^−1^
_[Fe+Mn+Zn]_) and superparamagnetism. Besides high *M*_s_, MTN also exhibits a magnetocaloric effect; the heating profile and corresponding SAR are shown in Figure 4B. It should be noted that we calculated the SAR value of MTN by data of the initial 4 min because a linear correlation was observed between time and temperature in the 4 min. However, the SAR in the research was lower than that of our previous study (SAR = 1102.4 W g^−1^) [18]. It was caused mostly by low *H*_applied_ (63.6 kA m^−1^), which was almost 55.4% of that (114.9 kA m^−1^) in our previous study. As low *H*_applied_ can improve biosafety of AMF, meanwhile, SAR of magnetic fluid is proportional to the square of *H*_applied_ [30], the low SAR value in the study is reasonable.

### 2.3. Drug Release Studies

To simulate intracellular conditions, an acidic buffer (pH = 5.0) containing acetate and glutathione (GSH, 1 mM) [31] was employed in drug release studies. As a control, other three types of buffers were also employed in drug release studies, including pure acidic buffer (pH = 5.0) without GSH, pure GSH buffer (1 mM), PBS with physiological pH value, and GSH (5 μM) [32]. The drug release profiles of MTN under different conditions were shown in Figure 5, indicating its high stability in the physiological environment and burst release behavior within cells.

As shown in Figure 5A, a small amount of Pt was detected from the outside solution of dialysis tubing in the physiological environment (PBS (pH = 7.4) + GSH (5 μM)) after 72 h incubation, which was consistent with previous studies [33,34]. On the contrary, MTN exhibited rapid release behavior in intracellular environment (pH = 5.0 + GSH (1 mM)). Under the intracellular niche, cumulative Pt release from MTN was rapidly within the initial 10 h, which was 7.27% at 1 h, then up to 63.67% at 10 h. To figure out the weighting of pH value and GSH concentration in platinum release, the Pt release profiles under acid environment (pH = 5.0) were contrasted with that under reductive environment (GSH, 1 mM), also shown in Figure 5A. It was clear that GSH could induce Pt release from MTN because of the high redox sensitivity of Pt(IV) [7]. However, in our study, the intracellular acid environment played a critical effect on triggering Pt release from MTN, as cumulative release in the acid condition is triple that in reductive condition at the same incubation time. The higher pH-sensitive release profile might be attributed to the number of ester bonding, which is triple of number of Pt(IV).

Furthermore, we studied the effect of MH on Pt release from MTN, as shown in Figure 5B. Apparently, MH of 20 min did not influence Pt release, as two release profiles with/without MH almost overlap at all time points. Therefore, the polymeric Pt(IV) is a typical macromolecular prodrug with an intracellular release profile, in which the acid environment is the most important factor involved in drug release. These results will benefit MTN application in vivo by enhancing intracellular release and reducing undesirable side effects.

### 2.4. Antitumor Efficacy In Vitro

Cytotoxicity experiments were studied by using 4T1. Before evaluation of the tumor inhibition ratio in vitro, biocompatibility of the mPEG-*b*-pHEMA was investigated and shown in Appendix A. As cell survival rates of mPEG-*b*-pHEMA at all concentrations (0.2–2 mg mL^−1^) approximated or even exceeded 100%, mPEG-*b*-pHEMA was a hydrophilic polymer with excellent biocompatibility.

Antitumor efficacy in vitro is shown in Figure 6, which displays various inhibition rates on 4T1 by different treatments.

Apparently, cisplatin exhibited a higher inhibition rate than that of its macromolecular prodrugs, PPM and MTN, for all tested periods (24 and 48 h). Meanwhile, the cytotoxicities of PPM and MTN on 4T1 were similar because they were prepared by the same macromolecular prodrug. Although periodic MH (20 min per 24 h) suppressed cell viability slightly at 24 h, it could enhance the antitumor efficacy of MTNs obviously. When incubation time reached 48 h, the cell viabilities of cisplatin and MTNs plus MH were almost overlapping at lower Pt concentration (≤5 μg mL^−1^).

In our previous study, we investigated the synergistic effect between MH and chemotherapy by the mediation of thermo-sensitive drug delivery, in which MH triggered chemotherapy release efficiently. However, in the study, the controlled release of cisplatin from polymeric Pt(IV) is independent beyond the effect of MH, as shown in Figure 4B. Moreover, periodic MH in the study showed limited cytotoxicity on 4T1, as cell viabilities of pure MH were 95.7% at 24 h and 89.1% at 48 h, both of which were lower than those of our previous study [19]. Therefore, the enhancement of polymeric Pt(IV) cytotoxicity by MH should be studied further.

By ICP-MS, we investigated intracellular Pt content after incubating with different formulations for 1 and 4 h, as shown in Figure 7.

First of all, formulation was a key factor in Pt internalization because cellular uptake of cisplatin showed higher Pt content at 1 and 4 h, in contrast with formulations of PPM and MTN. It is reasonable that cisplatin can enter cells via both passive diffusion and active uptake [35]. On the one hand, cisplatin is small enough to diffuse through cell membranes easily compared to PPM and MTN with larger sizes. On the other hand, copper transporter 1 (CTR-1) can mediate the active uptake of cisplatin efficiently. Because of the high expression of CTR-1 in the kidney proximal tubule [36], nephrotoxicity is the main side-effect of cisplatin in clinics. Unfortunately, 4T1 displayed high CTR-1 expression level (CTR-1/action ≥ 0.5) [37]. Consequently, cisplatin alone displayed efficient internalization (Figure 7) and high cytotoxicity (Figure 6).

Although MTN with the largest size distribution displayed the lowest level of Pt cellular internalization, MH enhanced cellular uptake of MTNs significantly at all time points (1 and 4 h). By stimulation of MH at an initial 20 min, we observed the highest Pt content among all groups at 1 h, which was higher than that of cisplatin. The phenomenon could be explained by thermal-induced increasing lipid fluidity and permeability of cell membranes. However, the influence of MH on MTN internalization is temporary. With prolonging culture time (4 h), Pt internalization in the group of MTN plus MH is still higher than that of MTN but lower than that of cisplatin significantly, as shown in Figure 7, corresponding to its cytotoxicity in Figure 6A. Even so, it is definite that MH is an efficient approach to increasing the cellular uptake of micellar Pt(IV). Therefore, the cytotoxicity of MTN should be improved further by periodic MH (20 min per 24 h) within prolonged culture time (48 h), corresponding to the result shown in Figure 6B.

### 2.5. Efficient Tumor Diagnosis by a Combination of MT and MRI In Vivo

It is well known that a multifunctional nanocomplex by combining noninvasive imaging and therapy represents the main development trend in tumor treatment because synchronous noninvasive monitoring possesses great advantages in visualizing drug biodistribution, assessing therapeutic responses, and predicting efficacy [38,39]. Among all common noninvasive imaging techniques, MRI is one of the most powerful diagnostic tools because of its high biosafety and remarkable spatial resolution. As nanocluster of SPIO was demonstrated its efficiency in shortening spin−spin relaxation time (T2) of MRI [40], in the study, MTN with a core of Mn_0.6_Zn_0.4_Fe_2_O_4_ nanocluster was used as a T2 contrast agent to locate tumors in vivo, as shown in Figure 8.

Because MTN has a suitable diameter of around 151 nm, MTN could distribute in solid tumors by EPR effect and induce slight darkening in a partial area of the tumor (Figure 8(A2)) compared to the counterpart without MTN injection (Figure 8(A1)). Unfortunately, the EPR effect is inadequate for nanocarrier accumulation in tumor sites, which is confirmed by many previous studies. As MTN exhibited a high *M*_s_ value and obvious magnet-induced distribution, as shown in Figure 4A, a button magnet was employed to improve MTN accumulation in tumors specifically. By attracting an external magnet nearby tumors, most areas of the tumor darkened visibly in Figure 8(B2), indicating the effectiveness of MT in increasing MTN biodistribution in tumors.

However, it should be noted that the accumulation of MTN in tumor sites did not equal its uniform distribution in tumor tissue. On the one hand, intrinsic barriers in tumors impeded the diffusion of MTN, as we mentioned previously in the introduction. On the other hand, a permanent magnet is inadequate for attracting MTN completely because the magnetic field of the permanent magnet will attenuate rapidly with increasing distance between tumor tissue and magnet. Meanwhile, the pH value of tumor tissue usually ranged from 6.5 to 6.8 [41], which was much higher than the effective pH value (≤5.0) in our drug release studies. It is necessary to promote penetration of MTN in tumor tissue and its endocytosis further. Considering the potential of MH on these aspects, we assessed the influence of MH on MTN distribution in tumors. However, the real-time temperature of tumors could not be monitored during MH in vivo because an overheated copper coil (≥50 °C) interfered with the detection of infrared radiation (IR) thermal camera seriously. The MRI results revealed diffusion of MTN after MH, as darkened areas of tumors in Figure 8(A3,B3) extended, by contrast with the corresponding counterpart before MH (Figure 8(A2,B2)). Especially for the group with MT, the region with T2 enhancement almost covered the whole tumor after MH. Considering high interstitial fluid pressure and dense extracellular matrix within tumors, passive diffusion of nanocarrier around 100 nm is almost impossible [42]. Therefore, we infer that local MH is the only driving force to facilitate the diffusion of MTN with a large diameter within the tumor.

To observe the biodistribution of MTNs directly, all major organs and tumors of mice were stained for iron detection after the MH. As ferric ion (Fe^3+^) can combine with ferrocyanide, resulting in a bright blue pigment, Prussian blue, we exhibited the distribution of MTN in different organs directly, as shown in Figure 9.

According to the results, MTN mainly accumulated in tumors and organs such as the liver and spleen after MH, which was consistent with a relative study [43]. In spite of the similar distribution in the liver between the two groups, MT improved the distribution of MTNs in tumors and reduced them in the spleen. As shown in Figure 9, in contrast with passive targeting (MTN alone), visible blue spots in tumors were more obvious in the group combined with MT. Meanwhile, Fe^3+^ content in the spleen showed an opposite trend. Besides the Fe^3+^ content, another marked significance between passive targeting and MT was the dispersion degree of these bright blue spots in tumors after MH. Compared to passive targeting, the group combining with MT exhibited an even distribution of MTN in the tumor section, which corresponded to MRI results after MH.

### 2.6. Antitumor Efficacy In Vivo

Based on the positive results we mentioned above, MTN not only showed low cytotoxicity alone and enhanced cytotoxicity and cellular internalization by combining with MH in vitro but also displayed MT enhanced tumor accumulation and MH stimulative tumor penetration in vivo. Encouraged by the excellent performance of MTN, we investigated the antitumor efficacy of MTN in vivo, especially under optimized strategy by integrating MT and MH, as displayed in Figure 10.

Apparently, the growth of the 4T1 tumor was very rapid, as tumor volume (*n* = 6) of the PBS group developed from 152 to 1008 mm^3^ in 15 days. After injection of different formulations with equivalent cisplatin content at 3 mg kg^−1^ body weight via intravenous injection, tumor growth was inhibited obviously, compared to that of the PBS group. Among these groups, MTN alone/MH did not display obvious tumor inhibition in the early stage because tumors in the early phase were too small to exhibit the EPR effect, inducing low accumulation of MTN in tumors and bad antitumor performance. When the tumor reaches a certain volume, low toxic MTN could accumulate in the tumor by the EPR effect and show a certain ability on suppressing tumor growth, as shown in Figure 10A. However, due to the limitation of the EPR effect, the accumulation of MTN was not enough to induce effective MH. The antitumor efficacies between MTN alone and MTN plus MH were similar in the whole study.

Owing to MT, the group of MTN plus MT exhibited more efficiency in suppressing tumor growth by enhancing the accumulation of MTN in tumors. With the repeating MT (4 h per 24 h), the performance of MTN plus MT was even better than that of cisplatin at the later stage. Although cisplatin inhibited tumor growth efficiently at an early stage, its antitumor ability declined rapidly after the third injection. Besides the treatments we discussed above, the group of MTN plus MT plus MH showed the strongest tumor suppression. The phenomenon should be ascribed to the effects of MT and MH on MTN efficacy, in which the former increased MTN accumulation in tumors and the latter improved MTN penetration in dense tumors. Moreover, because of the dual effect of chemotherapy and MH, the group of MTN plus MT plus MH did not only exhibit the smallest tumor size but also displayed a downtrend in shrinking tumor volume at the end of in vivo study, which indicates a possibility of tumor vanishment by prolonged treatment of MTN plus MT plus MH.

Furthermore, we calculated tumor growth inhibition (TGI) at the end of the antitumor study, which reflected the antitumor efficacies of different treatments directly. Corresponding to final tumor volumes, the TGIs of different treatments were 65.73% (cisplatin), 59.67% (MTN), 77.38% (MTN + MT), 62.29% (MTN + MH), and 88.38% (MTN + MT + MH), respectively. Apparently, the group of MTN plus MT plus MH showed the highest TGI up to 88.38%, which could be comparable to those of other cisplatin prodrug systems, such as poly-ermic Pt(IV) with optimal drug/copolymer ratio (75% of TGI) [44], polyermic Pt(IV) plus photothermal therapy (87.89% of TGI) [45], and polyermic Pt(IV) plus another chemotherapy drug (83.4% of TGI) [46]. Therefore, a combination of MTN, MT, and MH should be a promising strategy for polymeric Pt(IV) application in clinics.

Considering high systemic toxicity of cisplatin, a safety evaluation was carried out by weighting the body weight of each mouse after injection, as shown in Figure 10B. Apparently, cisplatin possessed obvious systemic toxicity, as the weight loss of mice could be observed after cisplatin injection. Meanwhile, other mice with MTN injection alone showed a tendency toward gaining body weight, illustrating the high biosafety of MTN. Except for MTN, other MTN-based treatments also exhibited low toxicity, as the mice treated by these strategies, including MTN + MT, MTN + MH, and MTN + MT + MH, showed similar fluctuations of body weights between 20 and 23 g, and all of them fell within the normal weight range of BALB/c mice.

## 3. Materials and Methods

### 3.1. Materials

mPEG with an average molecular weight of 5000 Da (mPEG_5K_), HEMA (97%), and SA were purchased from Sigma-Aldrich. DCC (98%), HOBt (97%), 4-Dimethylaminopyridine (DMAP, 99%), and EDC (98%) were purchased from Tokyo Chemical Industry (TCI, Japan). 2,2-Azobis(isobutyronitrile) (AIBN, 99%), anhydrous N,N-Dimethyl sulfoxide (DMSO, 99.9%), anhydrous N,N-Dimethylformamide (99.8%, DMF), cisplatin (c,c,t-[Pt(NH_3_)_2_Cl_2_], 99%), and GSH were purchased from Aladdin (China). 3-(4,5-dimethyl-thiazol-2-yl)-2,5-diphenyl tetrazoliumbromide (MTT) was purchased from Beyotime Biotech. Co., Ltd. (Shanghai, China). Iron(III) acetylacetonate [Fe(acac)_3_], manganese acetylacetonate [Mn(acac)_2_], and zinc(II) acetylacetonate [Zn(acac)_2_] were purchased from Alfa-Aesar. 1,2-Hexadecanediol (97%), oleic acid (technical grade, 90%), oleylamine (technical grade, 70%), and benzyl ether (98%) were purchased from Sigma-Aldrich.

Dialysis tubing, hydrogen peroxide (H_2_O_2_, 30%), dichloromethane (DCM), tetrahydrofuran (THF), diethyl ether, acetone, and 1,4-dioxane were purchased from Sinopharm Chemical Reagent Co., Ltd. (Shanghai, China).

AIBN was purified by recrystallization from ethyl alcohol. HEMA was purified by passing through basic alumina to remove monomethyl ether hydroquinone. DCM was desiccated by reflux with calcium hydride (CaH_2_) and purified by reduced pressure distillation. Other reagents were used as received. The water used in all experiments was deionized with a Millipore Milli-Qsystem.

The monodisperse magnetic nanoparticles Mn_0.6_Zn_0.4_Fe_2_O_4_ were synthesized following our previous study [18]. The typical synthetic procedure was described following. A certain amount of Fe(acac)_3_, Mn(acac)_2_, Zn(acac)_2_, 1,2-hexadecanediol, oleic acid, and oleylamine with molar ratios of 10/3/2/50/30/30 were dispersed successively in benzyl ether. After deoxidizing by argon at 100 °C over 30 min, the mixture was heated to 200 °C for 2 h under an argon atmosphere, then heated further to reflux (≈295 °C) for one and a half hours. The product, monodispersed Mn_0.6_Zn_0.4_Fe_2_O_4_ magnetic nanoparticles, was precipitated by excess ethanol, then collected by centrifugation. The purified process was repeated three times. Finally, the magnetic nanoparticles were dispersed in THF with a concentration of 10 mg mL^−1^ for storage under −20 °C.

### 3.2. Synthesis of Pt(IV) with Single Carboxyl

The preparation of Pt(IV) with a single carboxyl group (-COOH) was performed by oxidization of cisplatin and following reaction with SA, according to previously described [47]. Cisplatin (0.5 g, 1.67 mmol) was suspended in water (10 mL) by magnetic stirring, then excess H_2_O_2_ (30% *w*/*v*, 15 mL) was added to the reaction in dark conditions. The reaction was processed for 5 h at 70 °C with continuous magnetic stirring and dark conditions. The product, hydroxy-modified Pt(IV), named *c*,*c*,*t*-[Pt(NH_3_)_2_Cl_2_(OH)_2_] or Pt(IV)-(OH)_2_, was purified by rinsing with water, ethyl alcohol, and diethyl ether, respectively.

After desiccating in vacuum, Pt(IV)-(OH)_2_ (0.2 g, 0.6 mmol) was dissolved in anhydrous DMSO completely by magnetic stirring, then succinic anhydride (0.06 g, 0.6 mmol) was added to the reaction under nitrogen condition. Under ambient conditions, the mixture was stirred for 12 h. Then, DMSO was removed by vacuum distillation. The final product, Pt(IV) with a single carboxyl group, also named *c*,*c*,*t*-[Pt(NH_3_)_2_Cl_2_OH(OOCCH_2_CH_2_COOH)] or Pt(IV)-COOH, was extracted by precipitation in excess acetone. By rinsing with acetone and diethyl ether, respectively, Pt(IV)-COOH was purified and dried under a vacuum oven.

### 3.3. Preparation of Hydrophilic Block Copolymer mPEG-b-pHEMA

For preparing mPEG-*b*-pHEMA by RAFT reaction, we prepared mPEG-DDAT as macro-CTA firstly according to our established method [18] and showed the synthetic procedure as follows. A certain amount of mPEG_5k_ (1 g, 0.2 mmol), DDAT (0.22 g, 0.6 mmol), and DMAP (0.024 g, 0.2 mmol) were dissolved in anhydrous DCM (20 mL) completely by magnetic stirring. After that, DCC (0.123 g, 0.6 mmol) was added to the solution at 0 °C and under nitrogen conditions. The reaction was stirred for 30 min at 0 °C and 48 h at room temperature. After the reaction, the solution was concentrated and filtered to remove insoluble by-products. The resulting product was precipitated in an excess amount of diethyl ether to remove excess DDAT. In order to purify the product, the precipitate was reprecipitated other four times by the same process. Finally, the purified mPEG-DDAT was dried by freeze-drying and preserved in argon under low temperatures (−20 °C).

Using mPEG-DDAT as macro-CTA, we synthesized hydrophilic diblock copolymer mPEG-*b*-pHEMA by RAFT polymerization. Briefly, a certain amount of mPEG-DDAT and HEMA were dissolved in 1,4-dioxane, in which the feed molar ratio of mPEG-DDAT/HEMA was fixed at 1/50. Then, AIBN (10% of macro-CTA molar quantity) was added to the reaction as a catalyst. Under nitrogen conditions, the reaction was processed at 70 °C for 12 h and stopped by cooling down to 0 °C. The final product, mPEG-*b*-pHEMA, was purified by precipitation in excess diethyl ether three times and dried under vacuum.

### 3.4. Preparation of Polymeric Pt(IV)

The polymeric Pt(IV) was also prepared by esterification under the assistance of EDC and HOBt. Briefly, Pt(IV)-COOH (200 mg, 0.46 mmol), mPEG-*b*-pHEMA (76.7 mg, 0.23 mmol hydroxyl group), EDC (83.4 mg, 0.46 mmol), and HOBt (12.4 mg, 0.092 mmol) were dissolved into anhydrous DMF (10 mL) completely under magnetic stirring and nitrogen conditions. Then, the reaction was stirred under ambient conditions for the following 48 h. Thereafter, the solution was dialyzed directly against water to remove DMF, EDC, and HOBt. The lyophilized crude product was dispersed into DCM and centrifuged to remove insoluble by-products and excess Pt(IV)-COOH. Finally, the purified polymeric Pt(IV) was collected by repeated precipitation in an excess amount of diethyl ether and dried in a vacuum oven.

### 3.5. Characterization of Molecular Structures

The structure of small molecular prodrug Pt(IV)-COOH was characterized by FT-IR (Thermo Scientific, Nicolet iS50, Madison, WI, USA) and ^1^H NMR spectrum (Bruker, AVANCE II, Zurich, Switzerland). The molecular weight of Pt(IV)-COOH was measured by electrospray ionization mass spectrometer (ESI-MS), which was performed by a high-resolution time-of-flight mass spectrometer (MS, Bruker, MaXis, Karlsruhe, Germany) equipped with electrospray ionization (ESI). The composition and structure of the polymer were characterized by a ^1^H NMR spectrum with CDCl_3_ or DMSO-d6 as the solvent and analyzed by their chemical shifts relative to tetramethylsilane (TMS). The Pt content in polymeric Pt(IV) was determined by ICP-MS (Thermo scientific, Xseries II, USA).

### 3.6. Preparation of Polymeric Pt(IV) Micelles (PPMs) and MTNs

All micelles were prepared by self-assembly technology. For the preparation of PPM, polymeric Pt(IV) was dissolved in THF completely with the concentration of 5 mg mL^−1^. The mixed solution was then slowly added into excess deionized water under sonication and dialyzed against water for 48 h to remove THF and other impurities, in which a dialysis bag (8–14 KD) was used. The dialysis solution was purified by centrifugation (2000 rmp). After that, PPMs were collected from the supernatant by lyophilization and stored at −20 °C.

For the preparation of MTNs, a certain amount of magnetic nanoparticles Mn_0.6_Zn_0.4_Fe_2_O_4_ and polymeric Pt(IV) were dispersed in THF adequately with the same concentration as 3 mg mL^−1^. Then, the MTN was prepared by the same procedure.

### 3.7. General Properties of PPM and MTN

The morphologies of PPM and MTN were characterized by high-resolution transmission electron microscopy (HRTEM, JEOL, JEM-2100, Tokyo, Japan). Particle diameters of PPM and MTN were measured by DLS (Malvern, Nano ZS90, Worcestershire, UK). The content of Mn_0.6_Zn_0.4_Fe_2_O_4_ nanoparticles in MTNs was characterized by ICP-MS. The magnetic property of MTN was measured by a vibrating sample magnetometer (VSM, Lake Shore Cryotronics, Inc., LakeShore 7404, Westerville, OH, USA) at 300 K.

We also estimated particle size and colloidal stability of PPMs and MNTs by DLS after they dispersed in phosphate buffer solution (PBS) for 7 d. The concentration of these samples was fixed as 0.5 mg mL^−1^.

The magnetocaloric effect of MTNs was evaluated by calculating its SAR, according to our previous study [18]. MTNs were dispersed into water with a concentration of Mn_0.6_Zn_0.4_Fe_2_O_4_ as 0.1 mg mL^−1^, which was determined by ICP-MS. The colloidal solution (4 mL) was placed in AMF, which was generated by an alternating magnetic field generator (SPG-20AB, ShuangPing Tech. Ltd., Shenzhen, China). Then, the temperature change of the sample was recorded by a computer-attached fiber optic temperature sensor (FOT-M, FISO, Québec, Canada). Finally, the SAR was calculated by the formula described in our previous study [18].

In this part of the study, the frequency (*f*) and strength (*H*_applied_) of the AMF was fixed at 114 kHz and 63.6 kA m^−1^. The inner diameter of the heating coil was 20 mm.

### 3.8. Drug Release Studies

In a typical experiment, a certain amount of MTNs (20 mg) was dispersed into 2 mL of the acidic buffer. Then, the solution was placed into dialysis tubing (3 kDa) against 98 mL of the corresponding buffer. The process of drug release was performed at 37 °C in a sharking incubator. At every predetermined time interval, 1 mL sample was withdrawn from outside of the dialysis tubing and measured by ICP-MS to determine the content of Pt. At the same time, an equal volume corresponding buffer was added as a release medium. According to Pt content, the percentage of cumulative Pt release from the MTNs was calculated and determined finally by averaging three measurements.

Furthermore, we also investigated the influence of MH on drug release. In this part, the inner diameter of the heating coil was 40 mm, resulting in constant f (114 kHz) and reduced strength (15.9 kA m^−1^). Because of the limitation of the heating coil, the study was performed in a 50 mL centrifuge tube. Briefly, MTNs of 10 mg were dispersed into 1 mL of the acidic buffer with 1 mM GSH, followed by 49 mL of the corresponding buffer. To simulate conditions of actual MH in the research, 20 min MH per 1 h was applied and repeated 8 times in a typical study. During the study, the 50 mL centrifuge tube was placed in a sharking incubator at 37 °C after MH until the next MH. At every time interval of 0.5 h, a 0.5 mL sample was withdrawn from the centrifuge tube, and an equal volume of the corresponding buffer was added as a release medium. Finally, the percentage of cumulative Pt release from the MTNs was measured and calculated as we described above.

### 3.9. Cell Viability Assay In Vitro

The 4T1 was gifted by my colleague, Rui Mengjie, who purchased the cell line from the Chinese Academy of Sciences (Shanghai). 4T1 was cultured in RPMI-1640 medium (Gibco) containing 10% fetal bovine serum (FBS, Biological Industries, Israel), then placed in an incubator at 37 °C, 5% CO_2,_ and humidified circumstance.

The cell viability in vitro was performed by standard MTT assay in the study. Biocompatibility of magnetic nanoparticle Mn_0.6_Zn_0.4_Fe_2_O_4_ has been confirmed by our previous study. In the research, cytotoxicity of hydrophilic copolymer mPEG-*b*-pHEMA was studied and shown in supporting information (SI).

To evaluate the chemotherapy efficiency of PPMs and MTNs, 4T1 was seeded in 96-well plates at a density of 2 × 10^4^ cells per well in 100 μL of corresponding medium and incubated under standard culture conditions for 24 h. After that, the medium was replaced by a medium containing various drug formations of cisplatin, Pt(IV), PPMs, and MTNs, respectively, in which the final equivalent Pt concentration was varied from 0.625 to 20 μg mL^−1^. After incubating for 24 and 48 h, cell viabilities were investigated by MTT.

Furthermore, to assess the assistance of MH, 4T1 was seeded in a culture dish (35 mm) at a density of 2 × 10^5^ cells per dish in 2 mL of the corresponding medium. After incubating for 24 h, the culture medium was replaced by a medium containing MTNs (20, 10, 5, 2.5, 1.25, 0.625 μg mL^−1^). To make sure constant concentration of Mn_0.6_Zn_0.4_Fe_2_O_4_ (60 μg mL^−1^) in the study, we prepared additional magnetic nanocluster fluid according to our previous study [17], then complement shortage of SPIO in groups of Pt concentration as 10, 5, 2.5, 1.25, 0.625 μg mL^−1^. Afterward, cells were exposed to AMF (20 min per 24 h, 114 kHz, and 15.9 kA m^−1^) at the beginning of the 24, following culture in an incubator under standard culture conditions. When total culture time reached 24 and 48 h, cell viabilities were also quantified by MTT.

As the culture medium was selected as a negative control, cell survival rates of different treatments were calculated as the percentage of negative control values.

### 3.10. Cellular Uptake Studies

4T1 were also seeded in culture dish (35 mm) with density of 2 × 10^6^ cells per dish. After incubating for 24 h, cells were treated with cisplatin, PPM, MTN, and MTN plus MH (20 min), respectively. For each formulation, the equivalent Pt content was fixed at 2.5 μg mL^−1^ (2 mL); meanwhile, the SPIO concentration of MTN plus MH was fixed at 60 μg mL^−1^ (2 mL). The drug uptake of Pt was performed by incubating cells with the above-mentioned drug formulations for 1 and 4 h. The cellular uptake of MTN plus MH was treated by exposure to AMF (20 min, 114 kHz, and 15.9 kA m^−1^) firstly, then incubated under standard culture conditions for prolonged 40 and 220 min, respectively. After removing the supernatant and washing three times, the cells in each culture dish were digested and counted to collect 1 × 10^6^ cells. After repeated freezing and thawing, the intracellular Pt content was determined by ICP-MS.

### 3.11. Animal Protocol

BALB/c mice (female, 5–7 weeks old, 18–20 g weight) were purchased from the laboratory animal center of Jiangsu University and maintained in the laboratory animal center of Jiangsu University under specific pathogen-free conditions. The orthotopic breast tumor was established by injection of 4T1 cells (1 × 10^7^ cells per mL, 100 μL per mice) into the fourth mammary fat pads of mice. After eight days, the tumor can grow to 100 mm^3^. Animals were treated according to the ethical guidelines of Jiangsu University. The animal experiments (approval code: UJS-IACUC-2021092703) were carried out according to the regulations for animal experimentation issued by the State Committee of Science and Technology of the People’s Republic of China.

### 3.12. Biodistribution of MTNs and MRI of Tumor In Vivo

When tumors grew to 200–300 mm^3^, these mice were divided into two groups (*n* = 5): MTN alone and MTN plus MT. MRI studies were performed with a 3.0-T clinic MRI imaging system (Siemens Trio 3T MRI Scanner) by using a micro coil for transmission and reception of the signal. The T2-weighted images were acquired by these conditions, which was listed as following: TR = 5000 ms, TE = 10–90 ms, slice thickness = 3 mm, flip angle = 150°, matrix size = 256 × 256, FOV = 100 mm, echo length = 8. Before injection of MTN, tumor-bearing mice were scanned by MRI. Then, the MTN solution prepared by dissolving into PBS at a dose of 4.2 mg kg^−1^ (mFe + mMn + mZn) was injected with 0.1 mL MTN solution from tail-vein for each tumor-bearing mouse. For the group of MT, the button magnet with a surface magnetic intensity of 0.18 T (diameter of 10 mm and thickness of 4 mm) was placed on the tumor area after injection immediately and maintained for 4 h. At 20 h later, after injecting MTNs, an MRI scan was performed to observe the T2 signal of the tumor site.

After that, the influence of MH on the penetration of MTNs within the tumor was also observed by MRI. After the second scanning, mice were treated with MH (20 min of each mouse, 114 kHz, and 15.9 kA m^−1^), then scanned by MRI following. In reality, the last scanning was performed at 22 h after MTNs injection.

To ensure the distribution of MTNs in vivo, a histological examination was performed after the last MRI scanning. The main organs (heart, liver, spleen, lung, and kidney) and tumors were extracted and fixed in 10% formalin, following treatment with nuclear fast red and Prussian blue stain.

### 3.13. In Vivo Tumor Inhibition Studies

When tumors grew to 150 mm^3^, these mice were divided into 6 groups (*n* = 6), injected with PBS, cisplatin, MTN, MTN plus MT (MTN + MT), MTN plus MH (MTN + MH), and MTN plus MT plus MH (MTN + MT + MH), respectively, in which, the dosage of Pt was calculated as 3 mg kg^−1^ cisplatin. In these groups, the injection was performed every three days, and MH was operated under AMF (114 kHz and 15.9 kA m^−1^) for 20 min at 20 h after each injection. The MT was carried out as we described in the MRI study. During the treatment, we injected 5 times in each group. Tumor volume and body weight of tumor-bearing mice were measured and recorded every three days, in which tumor volume was calculated by a formula described in a relative study [43]. The TGI rates of different treatments were calculated to assess corresponding antitumor efficacies by a formula described in a relative study [48].

### 3.14. Statistical Analysis

First of all, all data were presented with mean ± standard deviation (SD). Secondly, a one-way analysis of variance (ANOVA) was used in the research to determine significant differences between pairs of two groups. *p* < 0.05 was considered statistically significant, and *p* < 0.01 was considered as significant extremely.

## 4. Conclusions

In the study, we developed a versatile MTN successfully with high stability, MH-facilitated accumulation, high-sensitivity MRI, MH-enhanced penetration, and efficacy of Pt(IV) by self-assembly of monodispersed Mn_0.6_Zn_0.4_Fe_2_O_4_ and amphiphilic polymeric Pt(IV). First of all, we synthesized macromolecular cisplatin prodrug with Pt content as high as 22.5% polymeric Pt(IV), then used it to prepare MTN by encapsulating superparamagnetic Mn_0.6_Zn_0.4_Fe_2_O_4_ nanoparticles. The MTN did not only exhibit the potential for passive targeting by combining a suitable diameter around 151 nm and high stability under physiological conditions but also displayed more abilities on MT, MH, and MRI because it possessed high *M*_s_ (103.1 emu g^−1^) and SAR (404 W g^−1^). More importantly, the drug release of MTN was sensitive to the intracellular environment, especially for low pH, indicating its low side effects in circulation. According to these properties, polymeric Pt(IV)-based formulations, PPM and MTN alone displayed low cytotoxicity, but the combination of MTN and MH showed as high cytotoxicity as cisplatin. After intravenous injection, targeted accumulation of MTN in tumors could be improved under MT, which facilitated observation of tumor by MRI. The further MH enhanced MTN penetration in tumors, which also could be reflected by MRI directly. By integrating MT and MH, MTN displayed the highest TGI at 88.38% and reduced the side effects of cisplatin simultaneously. Although we did not observe tumor extinction by cascade effects of MT, MH, and chemotherapy of polymeric Pt(IV), the MTN was still a competitive candidate for diagnosis and therapy of tumors in clinics because of its advantages in efficient targeting, high-sensitivity MRI, low toxicity, and high efficacy simultaneously.

## Figures and Tables

**Figure 1 pharmaceuticals-15-00480-f001:**
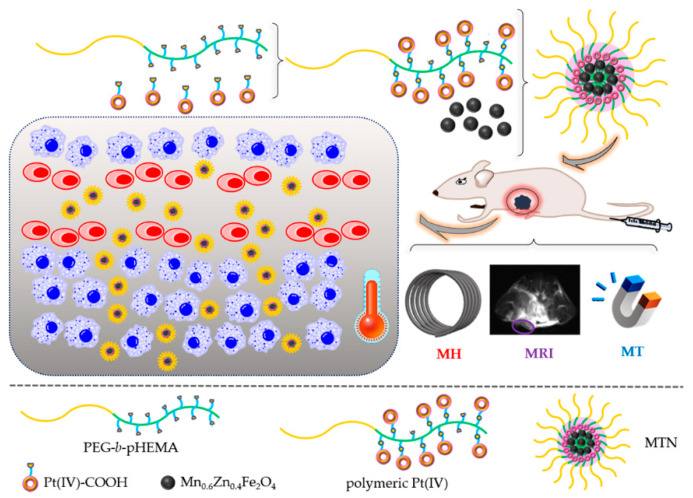
Schematic illustration of the theranostic nanocomplex formation and its cascading effects on diagnosis and therapy of tumors, including efficient magnetic targeting firstly, following magnetic resonance imaging, subsequent magnetic hyperthermia to promote penetration of nanocomplex, and enhanced antitumor efficiency of polymeric Pt(IV) finally.

**Figure 2 pharmaceuticals-15-00480-f002:**
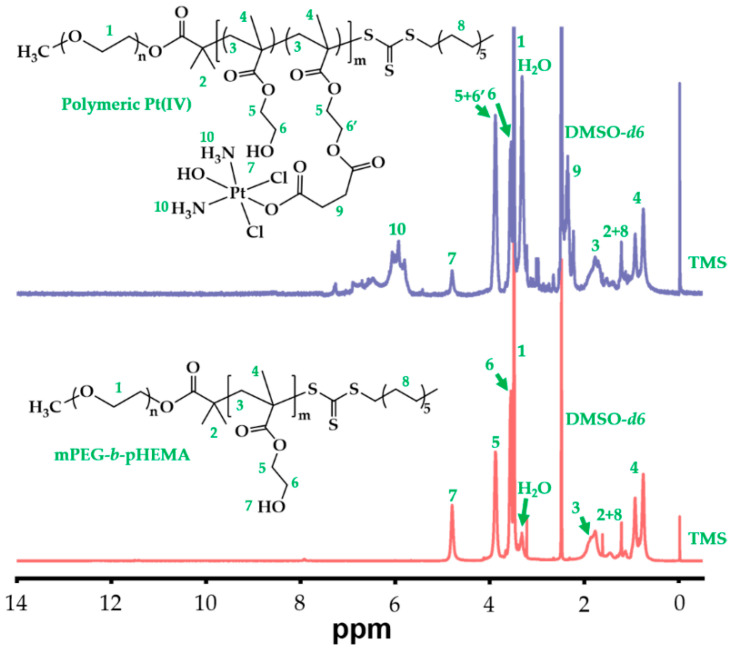
^1^H NMR spectra of mPEG-*b*-pHEMA and polymeric Pt(IV) in DMSO-*d6*.

**Figure 3 pharmaceuticals-15-00480-f003:**
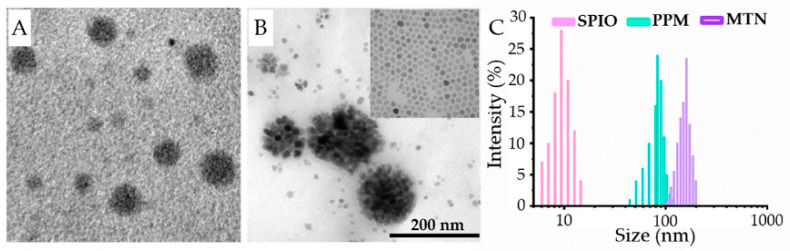
Morphologies (TEM) and particle size distributions (DLS) of PPM, MTN, and SPIO: (**A**) TEM result of PPMs; (**B**) TEM result of MTNs, inset: TEM result of SPIOs; (**C**) DLS results of PPM, MTN, and SPIO.

**Figure 4 pharmaceuticals-15-00480-f004:**
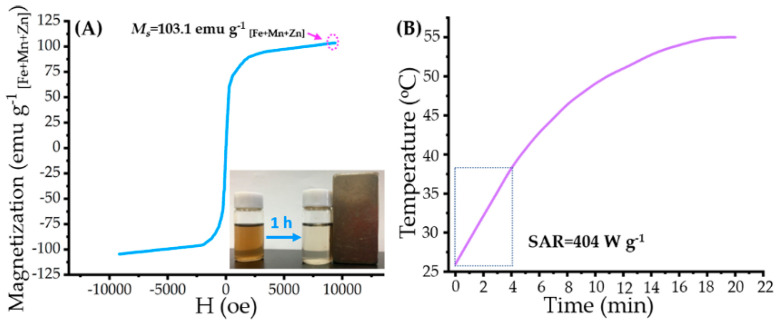
Magnetic property and magnetocaloric effect of MTNs: (**A**) magnetization curve of the MTN at 300 K; inset: photograph of MTN solution and its response to an external magnet at 1 h; (**B**) time-dependent temperature curve of MTN in AMF (corresponding parameters as 114 kHz of frequency and 63.6 kA m^−1^ of strength) and corresponding SAR value.

**Figure 5 pharmaceuticals-15-00480-f005:**
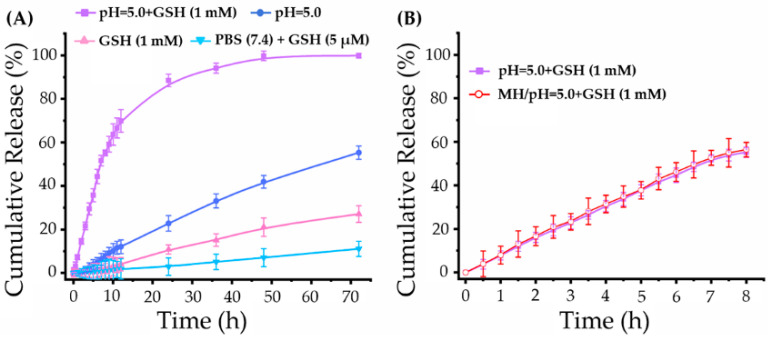
Drug release profiles of MTN under different conditions: (**A**) Pt release profiles of MTN under intracellular environment (pH = 5.0 + 1 mM GSH), physiological condition (pH = 7.4 + 5 μM GSH), pure acid environment (pH = 5.0), and pure reductive environment (1 mM GSH); (**B**) the influence of MH on Pt release under intracellular environment.

**Figure 6 pharmaceuticals-15-00480-f006:**
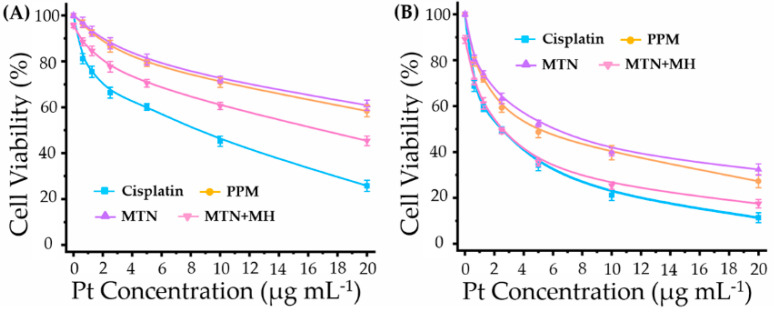
The cytotoxicity of 4T1 treated with cisplatin, PPM, MTN, and MTN + MH (duration time of MH: 20 min per 24 h) for 24 h (**A**) and 48 h (**B**).

**Figure 7 pharmaceuticals-15-00480-f007:**
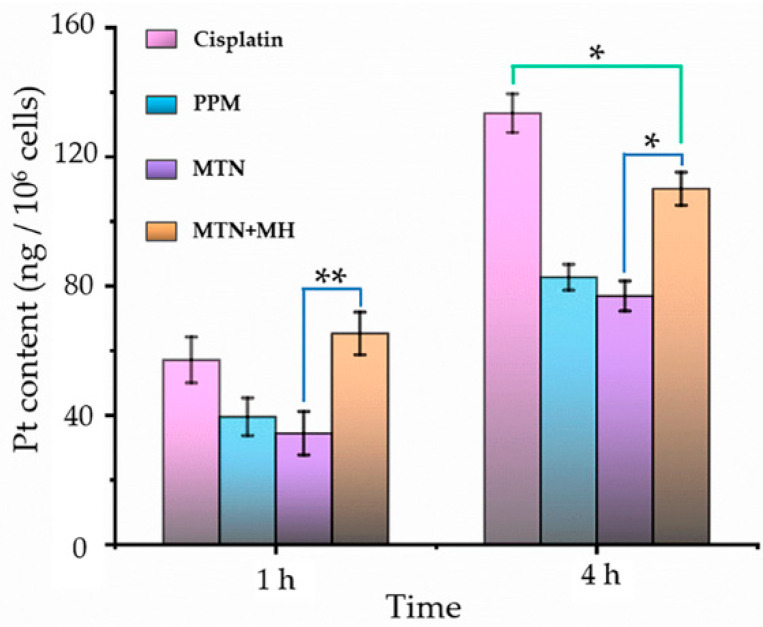
The intracellular Pt contents by treating with cisplatin, PPM, MTN, and MTN + MH for 1 h and 4 h. The MH was operated for the initial 20 min of the study. (* *p* < 0.05; ** *p* < 0.01).

**Figure 8 pharmaceuticals-15-00480-f008:**
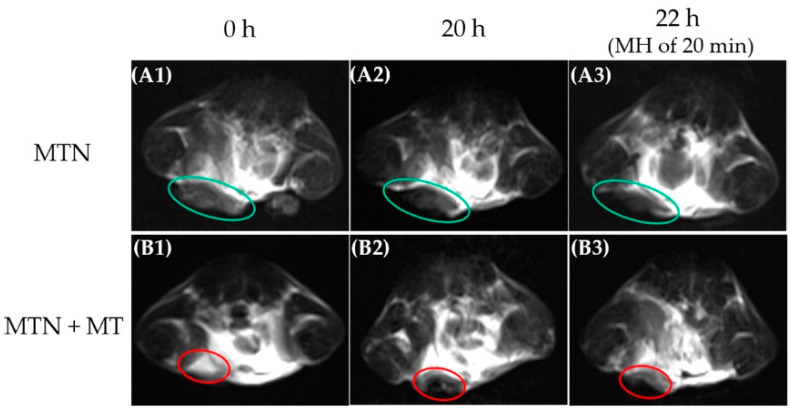
T2-weighted MRI results of orthotopic breast tumor before and after i.v. administration of MTN with corresponding treatments: MTN (**A1**–**A3**) and MTN + MT (**B1**–**B3**). Before intravenous injection of MTN, orthotopic breast tumors were scanned by MRI as control (**A1**,**B1**). After injection, the duration time of MT was 4 h, and the tumors were scanned again at 20 h (**A2**,**B2**). After MH of 20 min, the third MRI results (**A3**,**B3**) were obtained at 22 h after intravenous injection of MTN.

**Figure 9 pharmaceuticals-15-00480-f009:**
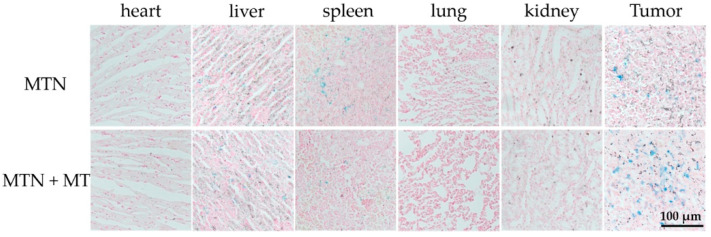
Tissue slices of main organs and tumors after treatments by MTN or MTN + MT. The duration time of MT was 4 h, and those tissues were obtained at 22 h after intravenous injection of MTN. These tissue slices were stained by nuclear fast red and Prussian blue simultaneously.

**Figure 10 pharmaceuticals-15-00480-f010:**
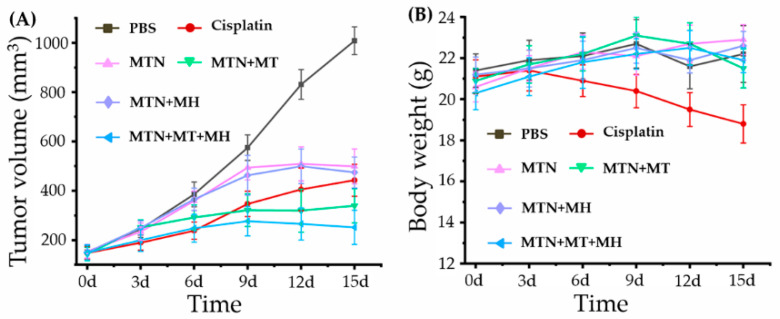
In vivo antitumor activities of different treatments: (**A**) the tumor volume curves after different treatments with the extension of curative time from 0 to 15 d; (**B**) the body weight curves after corresponding treatments with the extension of curative time from 0 to 15 d.

## Data Availability

Not applicable.

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
