# Peer review of "A Theranostic Nanocomplex Combining with Magnetic Hyperthermia for Enhanced Accumulation and Efficacy of pH-Triggering Polymeric Cisplatin(IV) Prodrugs"

_pharmaceuticals, 2022, doi:10.3390/ph15040480_

Round 1

Reviewer 1 Report

The paper is an interesting investigation about the synthesis and characterization of a theranostic nanocomplex for enhanced Pt(IV) anticancer activity. The topic of the paper is worthy of investigation and well fits with the scope of the journal. The overall evaluation of the paper is positive, and it can be published in Pharmaceuticals after addressing the following concerns:

  • Abstract

Some additional results data (if possible) should be inserted here.

  • Introduction

authors well presented their experimental idea, design, and results. This reviewer is suggesting to improve the presentation of the novelty statement of the study.

  • Results and Discussion

Figure 3b. Authors should elucidate if the analysis of mean diameter by DLS is number-weighted or intensity weighted.

Release Profile: authors should take into consideration the possibility to try fitting the release curves in order to better describe the linear profile in the first 10h and the subsequent jump (lines 233-234).

Anticancer activity: authors should consider the possibility to compare their results with different systems (if available) in the literature to strength the outcome of their study also in terms of efficiency

Author Response

Response to the First referee's comments

Comment 1: “Abstract: Some additional results data (if possible) should be inserted here.”

Answer: Thank you very much for your valuable advice, the abstract has been modified carefully by inserting more data to support our research. The relative details have been remarked by red color in the “version of revised manuscript for reviewers”. I hope that reviewer would like the new version of our manuscript.

 Comment 2: “Introduction: authors well presented their experimental idea, design, and results. This reviewer is suggesting to improve the presentation of the novelty statement of the study.”

Answer: Thank you very much for your approval. It is my pleasure to present the research. According to your valuable advice, I organized the whole study and modified the introduction thoroughly. The relative details have been remarked by red color in the “version of revised manuscript for reviewers”. I hope that reviewer would like the new version of our manuscript.

Comment 3: “Results and Discussion: Figure 3b. Authors should elucidate if the analysis of mean diameter by DLS is number-weighted or intensity weighted.”

Answer: Thank you very much to correct our negligence. Based on the comment, we modified the Figure 3b, adding captions of horizontal and vertical coordinates as “Size (nm)” and “Intensity (%)” respectively. I am really sorry for our negligence.

Comment 4: “Results and Discussion: Release Profile: authors should take into consideration the possibility to try fitting the release curves in order to better describe the linear profile in the first 10h and the subsequent jump (lines 233-234).”

Answer: Thank you very much for your attention on our research. According to your comment, we attempted to fit data of MTN under intracellular environment (pH=5.0 + 1 mM GSH). Unexpectedly, the fitting result showed the cumulative release of platinum from MTN under intracellular environment conformed the first-order kinetics (marked by red color), as shown in following table:

Mathematical model (equation)

a

b

r2

Zero order

Mt/M=a+b*x

25.98

3.86

0.6302

First-order

Mt= M(1-e-bx)

99.61

0.097

0.9970

Higuchi

Mt=a*x1/2+b

14.57

3.86

0.8846

Considering hydrolytic cleavage of ester bonding is nonlinear in vivo, because biodegradable mode of polyester is bulk degradation. Pt release from MTN under intracellular environment should be nonlinear. Therefore, the description of Pt release as “linear profile” is inaccurate. I have modified relative description. However, I won’t to discuss the fitting of the release data hastily. The interesting phenomenon should be a key point in my next study. Thanks for your advice again.

Comment 5: “Results and Discussion: Anticancer activity: authors should consider the possibility to compare their results with different systems (if available) in the literature to strength the outcome of their study also in terms of efficiency.”

Answer: Thanks very much for your valuable comment. Based on the comment, we quantified anticancer efficacies of different treatments in my research by calculating TGI (tumor growth inhibition) rate. After that, we summarized relative researches on cisplatin prodrug system alone and combining with other treatment or drug. By comparing the TGI, the group of MTN+MT+MH in our study showed a competitive antitumor efficacy, as its TGI was slightly higher than those of other researches. I wrote a paragraph to describe this in the revised manuscript. And we also marked them with red color in the “version of revised manuscript for reviewers”.

Reviewer 2 Report

The manuscript of Qu et al. reports the preparation of a polymeric multisystem based on magnetic nanoparticles and cisplatin(IV) prodrug for cancer theranostics.

The paper is well written and it is supported by a fine analysis of results. Anyway, for considering this manuscript for publication in Pharmaceuticals, some issues need to be addressed:

- line 96-113: this long paragraph should not be included in the Introduction section. It could be an initial statement for the Results section or, better, a Conclusion paragraph. But it describes the final achievements of the activities and should be removed from the Introduction.

- line 138-140: remove the template editorial instructions.

- figure 4: indicate also in the caption the experimental details for MH (frequency and field).

- line 204-207: according to the explanation of the Ms decay, the reader could understand that the Ms was normalized considering the overall mass of MNT. Probably, a different way to represent these results could be to normalize the Ms to the grams of iron (by ICP) of the two formulations, SPIO and MNT. The Ms should be similar for the two formulations.

- figure 5: indicate the global temperature obtained by MH application in the experiment reported in 5B

- line 412: a short sentence on the synthesis of the nanoparticles could be included in M&M 2.6 sub-section.

- in general, it’s really a pity the missing information for the tumor temperature in the in vivo hyperthermia. For the next studies, I suggest the use of a high-speed thermal camera and the liquid-cooling of the copper coil.

Author Response

Response to the second referee's comments

Comment 1: “line 96-113: this long paragraph should not be included in the Introduction section. It could be an initial statement for the Results section or, better, a Conclusion paragraph. But it describes the final achievements of the activities and should be removed from the Introduction.”

Answer: Thank you very much for your attention on our research. According to your comment, I removed the long paragraph “line 93-116” from introduction and organized the introduction and conclusion thoroughly. The relative details have been remarked by red color in the “version of revised manuscript for reviewers”. I hope that reviewer would like the new version of our manuscript.

 Comment 2: “line 138-140: remove the template editorial instructions.”

Answer: Thank you very much to correct our negligence. Based on the comment, I removed the editorial instructions. I am really sorry for our negligence.

Comment 3: “figure 4: indicate also in the caption the experimental details for MH (frequency and field).”

Answer: Thank you very much for your attention on our research. According to your comment, I supplemented experimental parameters of MH, including frequency and strength of alternative magnetic field in the corresponding caption. The relative details have been remarked by red color in the “version of revised manuscript for reviewers”. I hope that reviewer would like the new version of our manuscript. Thank you very much.

Comment 4: “- line 204-207: according to the explanation of the Ms decay, the reader could understand that the Ms was normalized considering the overall mass of MNT. Probably, a different way to represent these results could be to normalize the Ms to the grams of iron (by ICP) of the two formulations, SPIO and MNT. The Ms should be similar for the two formulations.”

Answer: Thanks so much for your attention on our research. According to your comment, I calculated saturation magnetism (Ms) of SPIO and MTN by normalizing metal content [Fe+Mn+Zn], because Mn0.6Zn0.4Fe2O4 was involved in our study. Just as your predicted, the Ms of SPIO (102.7 emu g-1 [Fe+Mn+Zn]) and MTN (103.1 emu g-1 [Fe+Mn+Zn]) were almost same. In order to describe the result logically, we modified the Figure 4(A) and relative sentences. Moreover, the magnetization curve of SPIO was removed from the main body of the manuscript and supplemented as Figure S8 in the revised manuscript. The relative details have been remarked by red color in the “version of revised manuscript for reviewers”. I hope that reviewer would like the new version of our manuscript. Thank you very much.

Comment 5: “figure 5: indicate the global temperature obtained by MH application in the experiment reported in 5B.”

Answer: Thank you very much to review our paper so carefully and point out a detail of our experiment. According to your comment, I really tried to monitor global temperature during drug release under MH. Unfortunately, the drug release studies were operated in the last summer. The current room temperature is lower obviously (20 oC) than that of summer. Therefore, the current global temperature during drug release under MH is so weird, which can not represent actual temperature, when we obtain data presenting in Figure 5(B). Even so, I really appreciate your valuable comment, I will improve our investigation in my further study.

Comment 6: “line 412: a short sentence on the synthesis of the nanoparticles could be included in M&M 2.6 sub-section.”

Answer: Thanks so much for your attention on our research. According to your comment, I modified the part of “3.1 Materials”, adding raw materials for Mn0.6Zn0.4Fe2O4 preparation and corresponding preparation process. The relative details have been remarked by red color in the “version of revised manuscript for reviewers”. I hope that reviewer would like the new version of our manuscript. Thank you very much.

Comment 7: “in general, it’s really a pity the missing information for the tumor temperature in the in vivo hyperthermia. For the next studies, I suggest the use of a high-speed thermal camera and the liquid-cooling of the copper coil.”

Answer: Thanks so much for your attention on our research. The comment is very useful. I will improve equipment to present more experimental data in my next research. Thank you very much again.

Reviewer 3 Report

In this study, Qu et al have presented a nanocomplex to enhance therapeutic effect. This is an interesting study and it has presented interesting results. After clearing up the ambiguities, I recommend it for acceptance at Pharmaceuticals. - English language needs revision for clarity. - In the abstract, the purpose and important results of the study should be provided. - Biological barriers are not well described, it is suggested in one paragraph to address the main barriers that nanocarrier faces in blood circulation and tumor penetration. The following studies are suggested in this regard . - Discuss further the level of acidity and distribution of acidity in the tumor. - In addition to increasing tumor blood supply, magnetic hyperthermia can also improve membrane permeability. Discuss the limitations of magnetic hyperthermia and the duration of application. - Discuss the limitations of magnetic targeting and its duration. - Novelty of the study is not well presented. Further details on this study and its importance in treatment need to be provided. - Figure 1 is vague and needs to be revised and added to the caption description. - Captions of all figures should include more details. - In Figure 6, cell death is much higher than what occurs in animal models, giving the reason for the discrepancy. - The conclusion section should include the most important results of the study and the importance of the results in the clinical stages should be emphasized. The limitations of the present study should also be presented. - The manuscript is not well organized. Some duplicate information is also mentioned. It is suggested that the journal standard be considered for the reconstruction of the manuscript structure.

Author Response

Response to the third referee's comments

Comment 1: “English language needs revision for clarity.”

Answer: Thank you very much for pointing out my deficiency in my English expression. According to your comment, the revised manuscript has been organized and modified thoroughly, especially in Abstract, Introduction, Conclusion, and part of result and discussion. The relative details have been remarked by red color in the “version of revised manuscript for reviewers”. I hope that reviewer would like the new version of our manuscript.

 Comment 2: “In the abstract, the purpose and important results of the study should be provided.”

Answer: Thank you very much for the comment. I modified Abstract carefully, some important results were added and remarked by red color. I hope that reviewer would like the new version of our manuscript.

Comment 3: “Biological barriers are not well described, it is suggested in one paragraph to address the main barriers that nanocarrier faces in blood circulation and tumor penetration. The following studies are suggested in this regard.”

Answer: Thank you very much for your attention on our research. According to your comment, I supplemented background on shortcoming of polymeric prodrug in the third paragraph of introduction. In the paragraph, I summarized a series of predicaments of polymeric prodrug in tumor, which showed highly associated with biological barriers. The relative description has been remarked by red color in the “version of revised manuscript for reviewers”. I hope that reviewer would like the new version of our manuscript. Thank you very much again.

Comment 4: “Discuss further the level of acidity and distribution of acidity in the tumor.”

Answer: Thanks so much for your attention on our research. Indeed, the level of acidity and distribution of acidity in the tumor are important for acid-sensitive drug release. According to your comment, I supplemented relative description at the part of “2.5. Efficient tumor diagnosis by a combination of MT and MRI in vivo”, which have been remarked by red color in the “version of revised manuscript for reviewers”. I hope that reviewer would like the new version of our manuscript. Thank you very much.

Comment 5: “In addition to increasing tumor blood supply, magnetic hyperthermia can also improve membrane permeability. Discuss the limitations of magnetic hyperthermia and the duration of application.”

Answer: Thank you very much to review our paper so carefully. Indeed, hyperthermia can improve membrane permeability, consequently, effective magnetic hyperthermia also shows similar potential. However, application of magnetic hyperthermia in clinic is still immaturity. In my opinion, the crucial issue is the distribution of SPIO in vivo and its actual content in tumor. Because nanocarrier with diameter from 10 nm to hundreds nanometer, always distribute in liver and spleen preferentially, liver and spleen would suffer hyperthermia seriously, when tumor site could be heated obviously by MH. However, in the study, limitations of magnetic hyperthermia and the duration of application were not our key point. In fact, to overcome the obstacle, we have designed another study to construct a polymer with pH/thermo-sensitive, which could be stranded in tumor under effects of relative low pH environment (6.5-7.0) and hyperthermia (40 oC) simultaneous. Therefore, in the manuscript, we just used MH, under assistance of MT, to benefit diffusion of MTN. The actual effect and limitation of MH will be studied carefully in our further study.

Comment 6: “Discuss the limitations of magnetic targeting and its duration.”

Answer: Thanks so much for your attention on our research. According to your comment, I supplemented relative description at the part of “2.5. Efficient tumor diagnosis by a combination of MT and MRI in vivo”, which have been remarked by red color in the “version of revised manuscript for reviewers”. I hope that reviewer would like the new version of our manuscript. Thank you very much again.

Comment 7: “Novelty of the study is not well presented. Further details on this study and its importance in treatment need to be provided.”

Answer: Thanks so much for your attention on our research. The comment is very useful. I have modified the revised manuscript thoroughly, especially in Introduction to present novelty of the study. The deficiency of the research should be considered carefully to guide our further researches. Thank you very much again.

Comment 8: “Figure 1 is vague and needs to be revised and added to the caption description.”

Answer: Thank you very much for your attention on our research. According to your comment, the caption of Figure 1 has been supplemented to describe integrated design of the study. The relative details have been remarked by red color in the “version of revised manuscript for reviewers”. I hope that reviewer would like the new version of our manuscript.

 Comment 9: “Captions of all figures should include more details.”

Answer: Thank you very much for your attention on our research. Based on the comment, all captions in the main body have been modified by adding more details. The relative details have been remarked by red color in the “version of revised manuscript for reviewers”. I hope that reviewer would like the new version of our manuscript.

Comment 10: “In Figure 6, cell death is much higher than what occurs in animal models, giving the reason for the discrepancy.”

Answer: Thank you very much for your attention on our research. In reality, antitumor efficacy of chemotherapy drug in vitro is always more efficient than its efficacy in vivo, because the key factor on drug efficacy in vivo is drug distribution. The antitumor efficacy of chemotherapy drug in vitro reflects its cytotoxicity, and higher cytotoxicity usually equals to the higher side-effect. Therefore, cisplatin showed the lowest IC50, indicating its high side-effect, meanwhile, PPM and MTN showed higher IC50, indicating their low side-effect. After intravenous injection, by assistance of MT, MTN showed advantage on accumulating polymeric Pt(IV) in tumor, while cisplatin accumulated in kidney preferentially. Consequently, antitumor efficacy of chemotherapy drug in vitro does not equal to its efficacy in vivo.

Comment 11: “The conclusion section should include the most important results of the study and the importance of the results in the clinical stages should be emphasized. The limitations of the present study should also be presented.”

Answer: Thanks so much for your attention on our research. According to your comment, the conclusion has been modified thoroughly. The relative details have been remarked by red color in the “version of revised manuscript for reviewers”. I hope that reviewer would like the new version of our manuscript. Thank you very much once again.

Comment 12: “The manuscript is not well organized. Some duplicate information is also mentioned. It is suggested that the journal standard be considered for the reconstruction of the manuscript structure.”

Answer: Thank you very much to review our paper so carefully. According to your comment, I have modified the revised manuscript thoroughly. The relative details have been remarked by red color in the “version of revised manuscript for reviewers”. I hope that reviewer would like the new version of our manuscript. Thank you very much once again.

Round 2

Reviewer 1 Report

Authors well addressed all comments. The manuscript can be published in its current form.

Author Response

Response to the First referee's comment

Comment : “Authors well addressed all comments. The manuscript can be published in its current form.”

Answer: Thank you so much, it is my great pleasure!

Reviewer 3 Report

The authors have responded well to the concerns. However, before acceptance, the necessary explanations should be provided in relation to the difference between the in vitro model and in vivo model in the manuscript.

Comment 10: “In Figure 6, cell death is much higher than what occurs in animal models, giving the reason for the discrepancy.”

Author Response

Response to the referee's comment

Comment: The authors have responded well to the concerns. However, before acceptance, the necessary explanations should be provided in relation to the difference between the in vitro model and in vivo model in the manuscript.

Comment 10: “In Figure 6, cell death is much higher than what occurs in animal models, giving the reason for the discrepancy”

Answer: Thank you very much for your attention on our research. It is an interesting phenomenon in our study that antitumor efficacies of cisplatin and macro-prodrug in vitro showed an opposite trend of their counterparts in vivo. According to the referee’s comment, I will discuss the phenomenon from three aspects.

First of all, it is well known that the most chemotherapy drugs lack selectivity between tumor cells and normal cells. And apparently, cisplatin is a classical antitumor drug without targeting ability on tumor cells. According to its antitumor mechanism, as we described in the introduction of our manuscript “Cisplatin, a classical platinum (Pt) drug with +2 valence (Pt(II)), is one of the most successful antitumor drugs against tumor cells by crosslinking DNA via coordination bond”, DNA is the targeting of cisplatin. Therefore, the accumulation of cisplatin within cell is directly linked to its toxicity [1].

Secondly, the cellular uptake of cisplatin is determined by copper membrane transporter1 (CTR 1) primarily, which is proved by Howell and his colleague [2]. Therefore, the most significant dose-limiting side-effect in vivo is nephrotoxicity, because kidney proximal tubule overexpresses CTR-1, resulting into cisplatin accumulation in kidney [3]. Moreover, the cell model in our study is 4T1, a cell line derived from murine breast tumor, which displays overexpression of CTR-1 [4]. Logically, cisplatin should exhibit high cytotoxicity on 4T1, because of its efficiency on cell internalization. Meanwhile, both of PPM and MTN in our study showed same outer layer, the PEG coating. PEGylation is a prevailing solution for protecting nanocarriers from quick clearance during blood circulation. However, PEGylation inhibits cellular uptake, causing significant loss of delivery activity, which is the so-called “PEG dilemma” [5]. Consequently, in vitro studies, cisplatin showed the greatest antitumor efficacy (Figure 6) and highest Pt content within cells at 4 h (Figure 7).

Actually, we have discussed the relative results in our manuscript “On another hand, copper transporter 1 (CTR-1) can mediate active uptake of cisplatin efficiently. Because of high expression of CTR-1 in kidney proximal tubule [36], nephrotoxicity is the main side-effect of cisplatin in clinic. Unfortunately, 4T1 displayed high CTR-1 expression level (CTR-1/action ≥ 0.5) [37]. Consequently, cisplatin alone displayed the efficient internalization (Figure 7) and high cytotoxicity (Figure 6)”.

Thirdly, please allow me to quote a sentence ---“No chemotherapeutic drug can be effective until it is delivered to its target site.” [6]

Logically, the journey begins with the injection of drug/drug delivery into bloodstream, and continues through stages of circulation, extravasation, accumulation, cellular internalization, and finishes after tumor cell apoptosis induced by them. Although cisplatin showed high cytotoxicity in vitro under constant drug concentration, numerous researches in clinic found that cisplatin accumulated preferentially in kidney, rather than other tissues (including tumor tissue) after intravenous injection. Therefore, antitumor efficacy of cisplatin in vitro does not equal to its efficacy in vivo. The antitumor efficacy of cisplatin in vitro reflects its cytotoxicity under constant drug concentration, and its cytotoxicity usually corresponds to its side-effect.

Compared to cisplatin, the MTN showed advantages on long circulation by PEGylation, and tumor accumulation by EPR. Even so, antitumor efficacy of MTN (TGI=59.67%) was still lower than that of cisplatin (TGI=65.73%), due to high cytotoxicity of cisplatin. However, under assistance of MT, accumulation of MTN in tumor has been enhanced, as shown in MRI results, the TGI of MTN+MT group has exceeded that of cisplatin. Moreover, after assistance of MT, penetration of MTN in tumor have been improved further by effective MH, which also has determined by accumulation of MTN in tumor. Therefore, the group of MTN+MT+MH showed the highest TGI as 88.38%, which was much higher than that of cisplatin.

According to the above reasons, I think that the phenomenon is reasonable. Furthermore, there are some literatures showing the similar phenomenon [6-10]. At last, I really appreciate your comment for bringing the interesting result to our attention. Some ideas are inspired by the comment. I will improve my study on polymeric prodrug in future. And thank you for your attention on our research again.

Reference:

[1] Drug Delivery Strategies for Platinum-Based Chemotherapy. ACS Nano 2017, 11, 8560−8578

[2] The Role of the Mammalian Copper Transporter 1 in the Cellular Accumulation of Platinum-Based Drugs. Mol Pharmacol. 2009, 75, 324-330

[3] An integrated view of cisplatin-induced nephrotoxicity and ototoxicity. Toxicol. Lett. 2015, 237, 219–227

[4] Theranostics of Malignant Melanoma with 64CuCl2. J Nucl Med. 2014, 55, 812-817

[5] A Multifunctional Envelope-type Nanodevice for Use in Nanomedicine: Concept and Applications. Acc Chem Res. 2012,17,1113-1121

[6] Odyssey of a cancer nanoparticle: From injection site to site of action. Nano Today, 2012, 7, 606-618

[7] A NIR light triggered disintegratable nanoplatform for enhanced penetration and chemotherapy in deep tumor tissues. Biomaterials 245 (2020) 119840

[8] pH-sensitive doxorubicin-conjugated prodrug micelles with charge-conversion for cancer therapy. Acta Biomaterialia 70 (2018) 186-196

[9] Versatile preparation of intracellular-acidity-sensitive oxime-linked polysaccharide-doxorubicin conjugate for malignancy therapeutic. Biomaterials 54 (2015) 72-86

[10] Catalase-loaded cisplatin-prodrug-constructed liposomes to overcome tumor hypoxia for enhanced chemo-radiotherapy of cancer. Biomaterials 138 (2017) 13-21